# The epidemiology and management of chronic osteomyelitis in pediatrics – A systematic review

**Joan L. Robinson**[1]\*, **Deema Gashgarey**[2], **Nourah Alruqaie**[2], **Liz Dennett**[3], **M. Elizabeth Pedersen**[4]

**1** Department of Pediatrics, University of Alberta, Edmonton, Alberta, Canada, **2** Department of Pediatrics, University of Toronto, Toronto, Ontario, Canada, **3** Geoffrey and Robyn Sperber Health Sciences Library, University of Alberta, Edmonton, Alberta, Canada, **4** Department of Orthopedic Surgery, University of Alberta, Edmonton, Alberta, Canada

\* jr3@ualberta.ca

## Abstract

### Objectives

Infection leading to necrosis of any bone can lead to chronic osteomyelitis (CO), sometimes resulting in permanent orthopedic sequelae. There are no published guidelines on the optimal management of adult or pediatric CO The objective of this study was to analyze published evidence for the epidemiology and management of pediatric CO.

### Methods

Inclusion criteria were studies of any design (minimum 2 patients) in any language that included patients with CO up to 17 years of age and described the epidemiology or management of CO. Ovid Medline(R) ALL, Embase (via Ovid), CINAHL Plus with Full Text (via EBSCOhost) and Scopus were screened Jan 1, 1989 to Feb 13, 2025. Quality assessment was based on the degree of bias if one were to use that study to make decisions about management of CO. Studies were divided into those from middle-high and high-income countries versus studies from lower income countries. Data were extracted on demographics, biomarkers, pathogens, treatments offered, recurrences and orthopedic sequelae.

### Results

There were 41 included studies – 26 from middle-high- and high-income countries (904 cases total) and 15 from lower income countries (975 cases total). All were observational and only 19 of the 41 studies reported 7 or 8 of the 8 items deemed essential to make decisions about management of CO. Definitions of CO varied markedly. Analyzing the 17 studies that included a minimum of 10 consecutive cases, 627 of 1073 cases (58%) occurred in males. In these 17 studies, the tibia

**Data availability statement:** Raw data are now included in the manuscript as supplementary file S2.

**Funding:** The author(s) received no specific funding for this work.

**Competing interests:** The authors have declared that no competing interests exist.

or femur accounted for 630 of 934 cases (67%). In 212 of 287 cases (74%) with a single pathogen reported, that pathogen was *Staphylococcus aureus*. There were no apparent differences in sex, bones involved or pathogens by country income level. Most cases (with the notable exception of those in recent case series from the United States) were managed with debridement. This was typically followed by sequential intravenous/per os (IV/ PO) antibiotics with almost no patients managed with PO antibiotics alone. Twelve case series reported use of local antibiotic delivery in addition to systemic antibiotics, but none of these studies had a control group. Studies were too heterogeneous in design to allow for data to be directly compared or combined. However, there was no obvious relationship between the route or duration of antimicrobials and the incidence of recurrences or orthopedic sequalae.

## Conclusion

There is a great need for high quality studies of all aspects of diagnosis and treatment of CO. Empiric coverage should target *S. aureus*. The evidence is poor quality, but there is no evidence that prolonged courses of antibiotics prevent recurrences.

## Introduction

Chronic osteomyelitis (CO) was recently characterized as "a protracted, often indolent disease process with [1] presence of a sequestrum and/or [2] relapse of infection in the same site (bone) weeks to years after apparently successful treatment of the initial infection in that site." [1] CO has been divided into five types: i) CO occurring post-acute hematogenous osteomyelitis (AHO), ii) primary hematogenous CO with no preceding AHO, iii) CO from a contiguous focus, iv) CO from orthopedic hardware and v) post-trauma CO [2]. A recent survey of pediatric infectious diseases physicians in the United States and Canada demonstrated "tremendous variability" in the management of CO [3].

Major barriers to studying CO are the lack of a uniform definition and the heterogeneity of clinical presentations and severity. Another barrier is that chronic non-bacterial osteomyelitis (CNO) is often initially confused with CO with clues being that CNO often involves the axial skeleton and sometimes eventually involves more than one bone.

Management options for CO include combinations of parenteral and oral antibiotics and surgical debridement with or without direct placement of antibiotics in bone. Removal of orthopedic hardware is considered optimal when CO is associated with previous orthopedic surgeries, especially if bony fusion has already occurred or if cure is not achieved with other options.

The objective of this review was to systematically review the literature on management and outcomes of pediatric CO and summarize the demographics, pathogens, treatments offered, and outcomes.

## Materials and methods

This was a systematic review of the interventions and outcomes of pediatric patients with CO. The primary outcome was the recurrence rate. We analyzed cases

separately in upper-middle- or high-income countries versus low or lower-middle income countries as patients in resource-poor countries often present with very advanced disease so would be predicted to have poorer outcomes.

This review was not registered.

## Inclusion criteria

Inclusion criteria were studies of any design in any language of CO (however the authors defined it) with or without orthopedic hardware that included minimum 2 patients up to 17 years of age.

## Exclusion criteria

Exclusion criteria were studies that:

1) included adults unless pediatric cases were reported separately or a minimum of 80% of cases were pediatric.

2) combined sub-acute osteomyelitis (using whatever definition the authors chose) and CO.

3) reported primarily radiographic findings or surgical techniques.

4) included primarily cases now considered to be non-infectious (chronic recurrent multifocal osteomyelitis or CNO or mandibular case series).

5) were published prior to 1989, an arbitrarily chosen year as studies prior to that appeared to mainly be poor quality.

## Search methodology

A health sciences librarian searched Ovid Medline(R) ALL, Embase (via Ovid), CINAHL Plus with Full Text (via EBSCO-host) and Scopus from Jan 1,1980 until Feb 13, 2025. The search combined the concepts of chronic osteomyelitis and children (S1 File). Single case reports and conference abstracts were excluded. The search was validated by checking that it included numerous seed articles the authors had previously identified. Results from all searches were downloaded to Covidence (Veritas Health Information, Melbourne Australia) where they were deduplicated. Reference lists of included articles and reviews were reviewed for additional studies. Two independent reviewers screened the title/abstracts according to the inclusion and exclusion criteria. Conflicts were resolved through discussion.

## Data extraction

Data were extracted by one reviewer, including demographics, the biomarkers erythrocyte sedimentation rate (ESR), C-reactive protein (CRP) and white blood cell count (WBC), pathogens, treatment and outcomes (recurrences or orthopedic sequalae) and entered into REDCap. Organisms isolated from bone or operative specimens were considered pathogens. Based on studies showing markedly discrepant results from sinus and bone cultures [4,5], organisms isolated from pus, sinuses or fistulas were not included.

## Data analysis

Case series were classified into those where patients lived in upper-middle- or high-income countries versus low or lower-middle income countries as determined by the World Bank [6].

To determine the distribution of sexes, bones involved and pathogens, data were combined from series with minimum 10 cases where it seemed likely that consecutive cases of CO of all bones were enrolled.

The initial plan was to perform a meta-analysis of outcomes but this was not conducted due to i) the heterogeneity of CO definitions ii) the fact that often minimal or no data were provided on the initial management of cases that recurred and iii) the markedly variable durations and completeness of follow-up.

Data are reported as per the PRISMA guidelines (S2 File).

## Quality Assessment

Two reviewers independently assessed each study and then reached consensus through discussion on answers to the following questions, modified from the NIH Study Quality Assessment Tools | NHLBI, NIH and JBI JBI Critical Appraisal Tools | JBI tools to fit the current review by assessing the degree of bias if one were to use that study to make decisions about management of CO:

1. Was there a case definition?

2. Were cases stated to be or presumed to be consecutive?

3. Was there clear reporting of sex, age, bones involved and pathogens?

4. Was the mean duration of antibiotic therapy reported?

5. Is it reported how many cases had surgery at initial diagnosis of CO?

6. Is it reported how many cases required further antibiotics or surgery after the initial intervention?

7. Was minimum 6 months follow-up for recurrence reported for over half of patients?

8. Was minimum 6 months follow-up for orthopedic sequelae reported for over half of patients?

Studies were excluded only if they had no affirmative answers.

## Results

### Search

The search yielded 1139 unique titles and abstracts of which 41 met the inclusion criteria (Fig 1). A case series labelled CO of the clavicle was excluded as most likely had CNO [7]. The Canavese study [8] was excluded as all patients appeared to be included in the Rousset study [9]. Studies by Yeargan [10] and Matzkin [11] were both included, recognizing that there may be overlap for tibial CO managed in Honolulu 1990–1998. Data reported by Stevenson [12] and Beckles [13] were combined as they reported the same patients. Data as entered into REDCap are provided in S3 File.

### Quality assessment

All the studies included were observational. Quality assessment is shown in Table 1 with the number of reported items out of 8 being 0 (n = 0), 1–3 (n = 7), 4–6 (n = 15) and 7 or 8 (n = 19).

### Demographics and diagnostic features

There were 26 case series from upper-middle or high income and 15 from low or low-middle income countries (Table 2) reporting a total of 904 and 975 cases, respectively. The definitions of CO varied markedly, requiring a minimum of 10 days to 6 months of a variety of signs and symptoms (Table 2). Three case series reported the percentage of all osteomyelitis cases that presented as CO: 86% in Ethiopia [17], 66% in Nigeria [26] and 54% in Fiji [25].

In studies where biomarkers were reported, ESR was elevated in 55–100% and CRP in 11–100% of cases; WBC count was usually normal (Table 2). The percentage of cases with positive blood cultures were reported in two studies: 7/343 (2%) (18), and 5/67 (7%) [2]; it is not reported how many patients in these studies had blood cultures performed.

There were 20 studies that reported minimum 10 presumed consecutive CO cases. Cases were male in 627 of 1073 (58%) cases – 244/ 452 (54%) in 6 studies from middle-high and high income countries [18,33,37,38,43,44] and 383/621

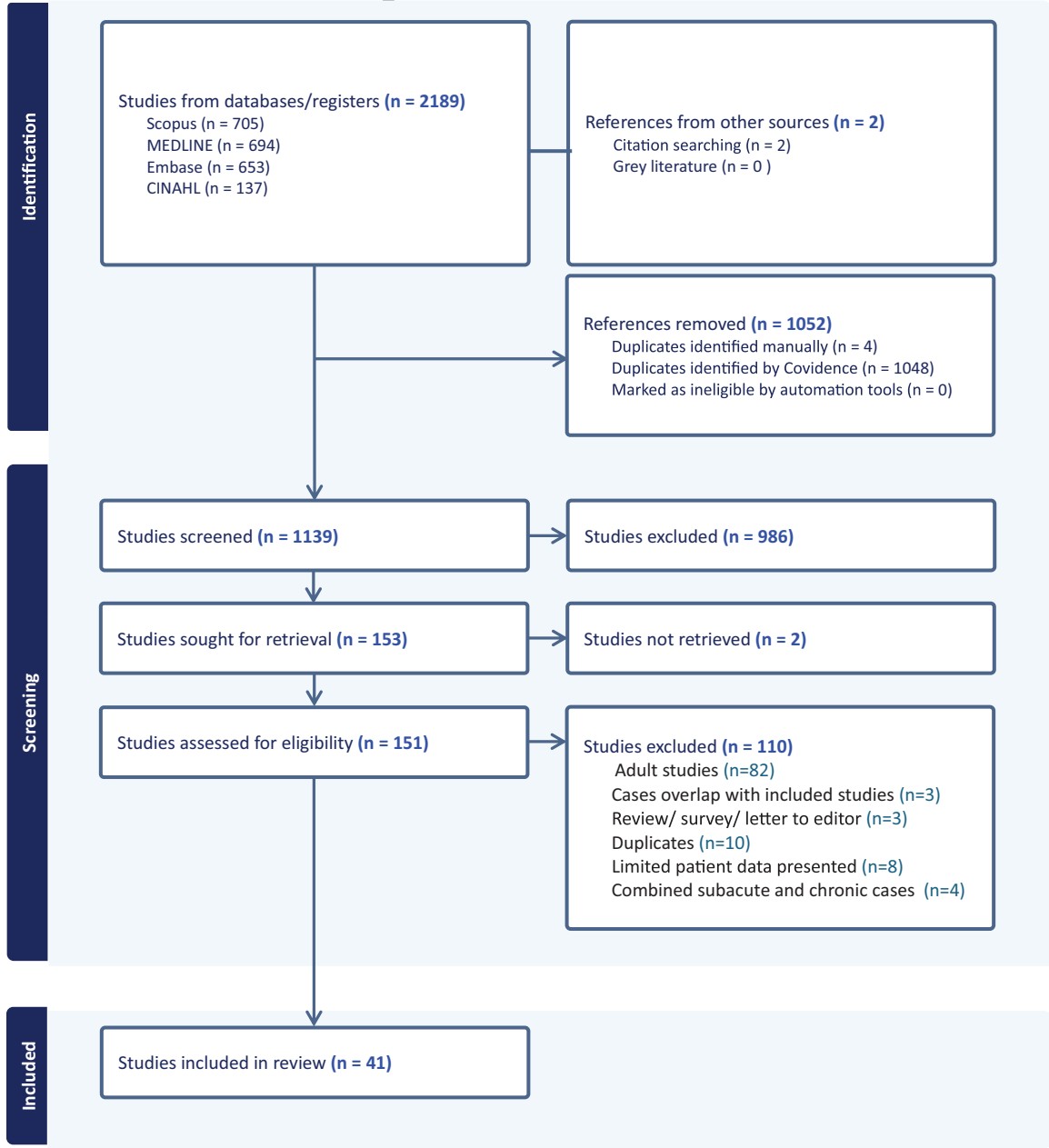

**Fig 1. PRISMA flow diagram of studies of chronic osteomyelitis in pediatric patients.**

(62%) in 11 studies from low and middle low income countries [11,12,15,22,23,26,30,32,41,45,47] (data missing for three studies [2,25,46]). The bones involved are shown in Fig 2 (data missing for 3 studies [18,26,46]), with 630 of 934 (67%) involving the tibia or femur; there are no apparent differences in the bones involved related to income level of country of origin. The pathogens were reported in 10 of these 20 studies as shown in Table 3. *Staphylococcus aureus* was isolated from 212 of 286 cases (74%) that reported a single pathogen – 77/115 (67%) in higher income countries and 135/171 (79%) in lower income countries. All other pathogens were isolated from 11 or fewer cases, even with all case series

**Table 1. Items reported for quality assessment of studies of pediatric chronic osteomyelitis with studies arranged by year – maximum score is 8.**

| Year | Author | Case definition | Cases presumed to be consecutive [1] | Sex, age, bones involved and pathogens | Antibiotics administered | Surgery at time of diagnosis | Need for further surgery after initial intervention | Minimum 6 months follow-up for orthopedic sequelae reported for over half of patients | Minimum 6 months follow-up for recurrence reported for over half of patients | Score |
|---|---|---|---|---|---|---|---|---|---|---|
| 2024 | Al-Alawi [14] | Y | Y | N | N | N | N | N | N | 2 |
| 2024 | Bhattacharyya [15] | N | Y | Y | Y | Y | Y | Y | Y | 7 |
| 2024 | Peshin [16] | Y | N | Y | Y | Y | Y | N | Y | 6 |
| 2023 | Mulualem [17] | Y | N | N | N | N | N | N | N | 1 |
| 2023 | Disch [18] | Y | Y | N | N | Y | N | N | N | 3 |
| 2023 | Shi [19] | Y | Y | Y | Y | Y | Y | Y | Y | 8 |
| 2022 | Lazzeri [20] | Y | Y | Y | Y | Y | Y | N | Y | 7 |
| 2021 | Kojima [21] | Y | Y | Y | N | Y | Y | Y | Y | 7 |
| 2021 | McNeil [2] | Y | Y | N | Y | Y | Y | N | Y | 6 |
| 2021 | Ellur [22] | Y | Y | Y | Y | Y | Y | Y | Y | 8 |
| 2019 | Edson [23] | Y | Y | Y | N | N | N | N | N | 3 |
| 2019 | Andreacchio [24] | N | Y | Y | Y | Y | Y | Y | Y | 7 |
| 2018 | Munshi [25] | N | Y | N | N | Y | N | N | Y | 4 |
| 2018 | Rousset [8] | Y | Y | Y | Y | Y | Y | Y | Y | 8 |
| 2018 | Omoke [26] | Y | Y | N | N | N | N | N | N | 2 |
| 2018 | Akyuz [27] | Y | N | Y | N | Y | Y | Y | Y | 6 |
| 2015 | Stevenson [12] – Beckles [13] | Y | Y | Y | Y | Y | Y | N | Y | 7 |
| 2015 | Costa [28] | N | N | Y | N | Y | Y | N | Y | 4 |
| 2015 | Shukrimi [29] | N | N | Y | Y | Y | Y | Y | Y | 6 |
| 2014 | Wirbel [30] | Y | Y | Y | Y | Y | Y | Y | Y | 8 |
| 2013 | Ponio [31] | Y | N | Y | N | Y | Y | N | N | 4 |
| 2011 | Mantero [32] | Y | Y | Y | Y | Y | Y | Y | Y | 8 |
| 2011 | Ulug [33] | Y | Y | Y | N | N | N | N | N | 3 |
| 2010 | Bar-On [34] | N | Y | Y | Y | Y | Y | Y | Y | 7 |
| 2010 | Zeng [35] | Y | Y | Y | Y | Y | Y | Y | Y | 8 |
| 2009 | Akakpo-Numado [36] | Y | Y | Y | N | N | N | N | N | 3 |
| 2008 | Dieckmann [37] | Y | Y | Y | Y | Y | Y | Y | Y | 8 |
| 2006 | Unal [38] | N | Y | Y | N | Y | Y | N | Y | 7 |
| 2005 | Matzkin [10] | N | Y | Y | Y | Y | Y | N | N | 5 |
| 2005 | Beslikas [39] | N | Y | Y | Y | Y | Y | Y | Y | 7 |

*(Continued)*

**Table 1.** (Continued)

| Year | Author | Case definition | Cases presumed to be consecutive [1] | Sex, age, bones involved and pathogens | Antibiotics administered | Surgery at time of diagnosis | Need for further surgery after initial intervention | Minimum 6 months follow-up for orthopedic sequelae reported for over half of patients | Minimum 6 months follow-up for recurrence reported for over half of patients | Score |
|------|--------|-----------------|--------------------------------------|----------------------------------------|--------------------------|------------------------------|-----------------------------------------------------|---------------------------------------------------------------------------------------|-------------------------------------------------------------------------------|-------|
| 2004 | Yeargan [9] | N | Y | Y | Y | Y | Y | Y | Y | 7 |
| 2002 | Paley [40] | N | Y | Y | N | Y | Y | Y | Y | 6 |
| 2002 | Bahebeck [41] | N | Y | Y | N | Y | Y | N | Y | 5 |
| 2001 | Rasool [42] | N | Y | Y | N | Y | Y | Y | Y | 6 |
| 2000 | Reinehr [43] | Y | Y | N | Y | Y | Y | Y | Y | 7 |
| 1997 | Vogely [44] | N | Y | Y | Y | Y | Y | Y | Y | 7 |
| 1995 | Bassey [45] | Y | Y | N | N | Y | Y | Y | Y | 6 |
| 1994 | Lauschke [46] | Y | Y | Y | Y | Y | Y | Y | Y | 8 |
| 1991 | Onuba [47] | N | Y | Y | Y | Y | N | N | N | 4 |
| 1991 | Tudisco [48] | Y | N | Y | N | Y | N | Y | Y | 5 |
| 1989 | Saïghi Bouaouina [49] | N | Y | Y | N | Y | Y | Y | Y | 6 |

[1] It was presumed that cases were consecutive even if this was not stated if it seemed likely that all cases during the study period were included. For studies of only one bone, cases were considered to be consecutive if all cases with that bone were presumably enrolled.

**Table 2. Diagnostic features of case series of pediatric chronic osteomyelitis arranged by year and country economy.**

| | Upper-middle and high-income Countries | | | | | | |
|---|---|---|---|---|---|---|---|
| Year | Author | Country | N | Definition of CO | ESR (mm/hr) | CRP (mg/L) | WBC (X 10⁹/L) |
| 2024 | Al-alawi [14] | Oman | 5 | persistence or recurrence of attributable symptoms and signs associated with a sequestrum, involucrum or osteosclerosis on a plain radiograph, requiring antibiotics for at least 12 weeks | | | |
| 2023 | Disch [18] | US | 343 | discharge diagnosis code for CO | | | |
| 2023 | Shi [19] [1] | China | 21 | confirmed by clinical features and imaging (plain radiographs, CT, and MRI) | | | |
| 2022 | Lazzeri [20] | Italy | 4 | confirmed by MRI (specific criteria NR) | | | |
| 2021 | Kojima [21] | Brazil | 5 | drainage from a fistula for at least 2 months | | 11.3, 12.8, NR (N=3) | |
| 2021 | McNeil [2] | US | 114 | (1) symptoms suggestive of osteomyelitis (e.g., pain, swelling, warmth, erythema, drainage, loss of function, etc.) for ≥28 days on presentation or (2) clearly documented history of acute osteomyelitis in a patient who received at least 4 weeks of effective antimicrobial therapy along with (a) new or worsening drainage, swelling, erythema, pain or loss of function; (b) radiographic evidence of sequestrum or permeative lucencies; or (c) readmission for the management of osteomyelitis | mean 28; range 10–60 | median 13 (IQR 5–33) | median 8.8 (IQR 6.9–12.2) |
| 2019 | Andreacchio [24] | Italy | 12 | NR | 20-30 (N=2); >30 (N=3); NR (N=7) | elevated in 42% | 17% elevated |
| 2018 | Munshi [25] | Fiji | 118 | NR | | | |
| 2018 | Rousset [9] | France | 8 | based on imaging but criteria NR – all had infected non-union | 37,64,77, NR (N=5) | normal (N=2), 37, 38 (N=2), 90, 127, >96 | 25% elevated |
| 2018 | Akyuz [27] [2] | Turkey | 3 | based on computed tomography examination of patients with sternocutaneous fistula | | | |
| 2015 | Costa [28] [3] | Portugal | 2 | NR | 10, 40 | normal (N=2) | normal (N=2) |
| 2015 | Shukrimi [29] | Malaysia | 3 | NR | 90, 95, NR | 30, NR, NR | 16, NR, NR |
| 2011 | Ulug [33] [4] | Turkey | 21 | had not improved clinically or microbiologically after ≥10 days of evolution, independent of the presence or absence of surgical and/or antimicrobial therapy | mean 72; range 8–125 | mean, 135.4±84.4 mg/ dl: range, 11–295 mg/dl⁵ | 48% elevated |
| 2010 | Bar-On [34] | Israel | 4 | NR | 48, 80, 117, NR | normal, normal, 9.4, 13.2 | |
| 2010 | Zeng [35] [6] | China | 2 | based on clinical findings and histopathology | | | |
| 2008 | Dieckmann [37] | Germany | 40 | based on histopathology | | | |
| 2006 | Unal [38] | Turkey | 22 | NR | all elevated | all elevated | |
| 2005 | Beslikas [39] [7] | Greece | 5 | NR | range 52–78 | range 2.5–12.6 | |
| 2004 | Yeargan [9] [8] | US | 30 | NR | | | |
| 2002 | Paley [40] [9] | US | 4 | NR | | | |
| 2001 | Rasool [42] [10] | South Africa | 10 | NR | | | |
| 2000 | Reinehr [43] | Germany | 10 | slight localized pain and/or swelling for minimum 2 weeks | 20-30 (N=3); >30 (N=3); NR (N=4) | 8.7, 13, normal (N=8) | |

*(Continued)*

| | **Upper-middle and high-income Countries** | | | | | | |
|---|---|---|---|---|---|---|---|
| Year | Author | Country | N | Definition of CO | ESR (mm/hr) | CRP (mg/L) | WBC (X 10⁹/L) |
| 1997 | Vogely [44] | Netherlands | 16 | NR | mean 24 | mean 33 | mean 10 |
| 1994 | Lauschke [46] [11] | Namibia | 30 | symptoms > 6 days with fever, elevated WBC count, pain and swelling | | | mean 10.0 (range 4.5–22.8) |
| 1991 | Tudisco [48] | Italy | 26 | reference to definition used in Tachdjiian chapter in 1990 edition of textbook *Pediatric Orthopedics* | | | |
| 1989 | Saïghi Bouaouina [49] | Algeria | 46 | NR | 91% elevated | | |
| | **Low and lower-middle income countries** | | | | | | |
| Year | Author | Country | N | Definition of CO | ESR (mm/hr) | CRP (mg/L) | WBC (X 10⁹/L) |
| 2024 | Bhattacharyya [15] | India | 10 | included all cases treated with calcium sulfate beads but no CO definition | mean 51; range 25–71 mm/L | mean 13; range 1–37 | mean 7.76; range 6.4311.07 |
| 2024 | Peshin [16] | India | 100 | pus discharge from an extremity persisting for more than 6 weeks with compatible radiological features | median 42 (IQR 25–54) | median 4.12 (IQR 1.45–11.52) | median 11 (IQR 9–14) |
| 2023 | Mulualem [17] | Ethiopia | 151 | 6 weeks of clinical signs and evidence of Brodie abscess or one or more of the following radiological findings: extensive sclerosis, sequestrum, involucrum, soft tissue swelling that obliterates the fat planes, peri-osteal reaction, lytic destructions, and cloaca | | | |
| 2021 | Ellur [22] [12] | India | 31 | NR | | | |
| 2019 | Edson [23] | Uganda | 75 | relapsing and persistent osteomyelitis characterized by low grade inflammation, presence of sequestrum, involucrum, Brodie abscess and fistulous tracts | | | |
| 2018 | Omoke [26] | Nigeria | 50 | infection lasting >6 weeks with radiological evidence of sequestrum, sclerosis or osteomyelitis associated with foreign bodies | mean 67.6 | | |
| 2015 | Stevenson [12]; Beckles [13] [12] | Malawi | 167 | Beit CURE Classification | | | |
| 2014 | Wirbel [30] | Afghanistan/ Angola (surgery in Germany) | 27 | duration > 6 months | | 11% elevated | |
| 2013 | Ponio [31] | Philippines | 80 | symptoms > 3 weeks with radiologic findings of seques-tration, bone destruction and cloaca formation | 55% of those measured elevated (31% not measured) | 30% of those measured elevated; 45% not measured | 29% elevated |
| 2011 | Mantero [32] | Kenya | 96 | symptoms for at least 6 months with fistula tract, and radiological evidence of sequestrum | | | |
| 2009 | Akakpo-Numado [36] | Togo | 23 | sequestrum and/or fistula | | | |
| 2005 | Matzkin [11] | Pacific Islands (sur-gery in US) | 55 | NR | 92% elevated; mean 53.5; range 8–130 | mean 26; range 10–168; elevated in 41% | median 7.9 (range 5.3–16.7) |
| 2002 | Bahebeck [41] [13] | Cameroon | 49 | NR | | | |

*(Continued)*

**Table 2.** (Continued)

| | Upper-middle and high-income Countries | | | | | | |
|---|---|---|---|---|---|---|---|
| Year | Author | Country | N | Definition of CO | ESR (mm/hr) | CRP (mg/L) | WBC (X 10⁹/L) |
| 1995 | Bassey [45] | Nigeria | 41 | sequestra and new bone formation, Brodie abscesses and bone sclerosis | | | |
| 1991 | Onuba [47] | Zimbabwe | 20 | NR | | | |

Legend: CO – chronic osteomyelitis; NR – nor reported: US – United States.

[1]Only long bone cases included.

[2]Only sternal cases included.

[3]Only Q fever cases included.

[4]Only cases with sinus tracts were included as the primary outcome was comparison of sinus and bone cultures.

[5]Units quoted in manuscript appear to be incorrect.

[6]Only orbital cases included.

[7]Only pelvic cases included.

[8]Only tibial cases included.

[9]Only cases with infected intramedullary nails included.

[10]Only calcaneal cases included.

[11]Only hematogenous cases that required surgery included.

[12]Only hematogenous cases included.

[13]Only cases that required surgery included.

combined. McNeil is the only study that analyzed pathogens by type of CO. Post AHO CO was almost always due to *S. aureus*, but for all other types of CO, *S. aureus* accounted for a minority of cases [2].

### Treatment and outcomes

All cases received antibiotics except for three cases managed with multiple surgeries [10] and one case ultimately labelled chronic multifocal osteomyelitis (Table 4). Patients managed with per os (PO) antibiotics alone were limited to two with Q fever [28] and possibly some of the 167 reported by Beckles [13] (cases with sclerosis were treated with 6 weeks PO flucloxacillin but it is not clear how other cases were managed). All cases in all other case series received intravenous (IV) antibiotics, usually followed by PO antibiotics for widely variable durations. Few studies provided complete data on choice of antibiotics.

In addition to systemic antibiotics, vancomycin [19,21], tobramycin [24] or gentamicin [9,34,37,44] alone or in combination [15,22,40] or unspecified antibiotics [13] were directly implanted into bone via cement, polymethylmethacrylate (PMMA) or calcium sulphate at initial debridement in 12 studies from 1997 to 2023 for 4–40 patients (mean 14) (Table 4). A study with vancomycin also used bioactive glass [21] (which has antibacterial properties) while another used bioactive glass alone [20,21]. No studies had a control group that received only systemic antibiotics. All report initial success, but one that used gentamicin reported that 6 of 40 had recurrences at 23 days to 3.5 years with one having a second recurrence 9 months later [37].

In most studies, all patients had at least one surgery with the notable exception of two recent United States (US) series where 34 of 114 (30%) [2] and 280 of 343 (82%) [18] were managed with antibiotics alone.

Table 4 shows the incidence of recurrences and orthopedic sequelae. The highest rates of recurrence were 26% in studies from the US [2] and from Fiji [25]. Typically, minimal data were provided on the characteristics or prior therapy of those with recurrences so this could not be further analyzed. A statistical comparison of outcomes was not conducted

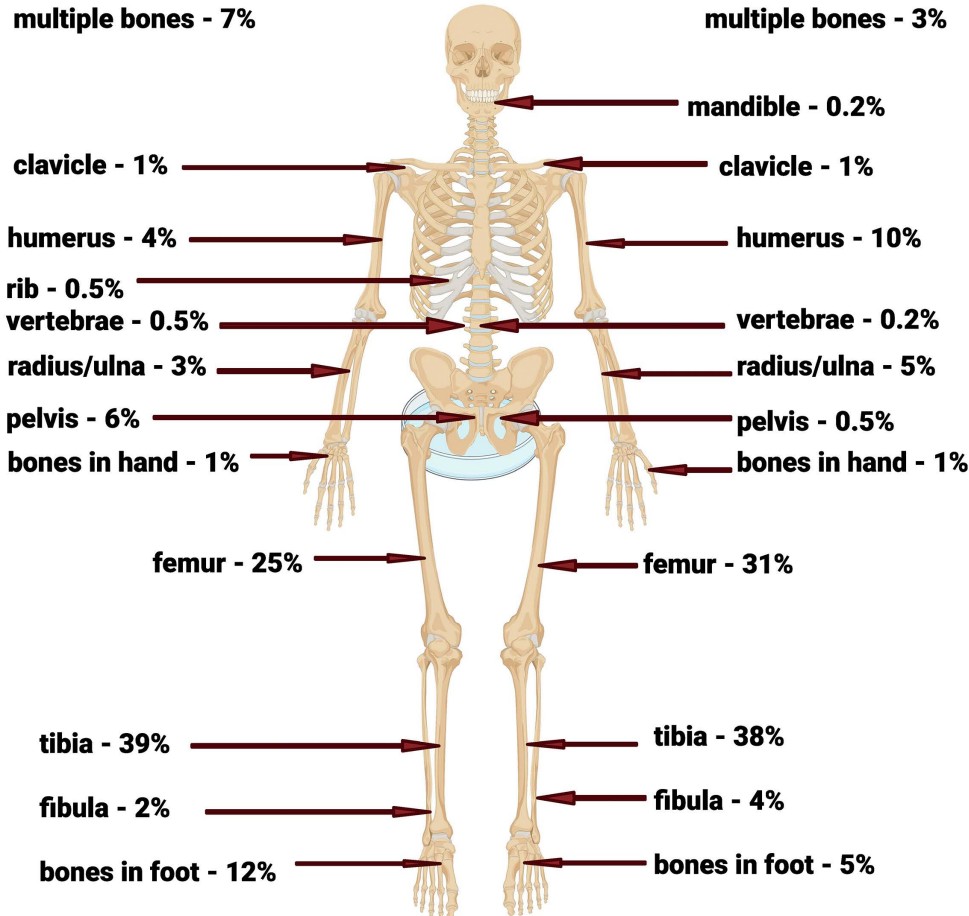

**Fig 2. Bones involved in pediatric chronic osteomyelitis – Percentages on the left are from 330 patients in 7 studies in middle-high to high income countries.** Percentages on the right are from 604 cases in 10 studies from low and low-middle income countries. Fig created by Biorender.

given the heterogeneity of definitions and incomplete descriptions of management, but there is no obvious link between the duration or route of delivery of antibiotics and outcomes.

## Discussion

This review summarizes 41 studies of pediatric CO. All were case series. There was no other study types identified. Sixteen did not provide a definition of CO. Each of the other 25 studies applied a unique definition. A Brodie abscess, sequestrum, or involucrum are proof of CO and are sometimes apparent on imaging. Other times the diagnosis is based on the presence of a sinus tract or on recurrence following completion of treatment for AHO. From the 41 studies, the McNeil definition would appear to be the most comprehensive: " [1] symptoms suggestive of osteomyelitis (e.g., pain, swelling, warmth, erythema, drainage, loss of function, etc.) lasted ≥28 days on presentation or [2] there was a clearly documented history of acute osteomyelitis in a patient who received at least 4 weeks of effective antimicrobial therapy along with (a) new or worsening drainage, swelling, erythema, pain or loss of function; (b) radiographic evidence of sequestrum or permeative lucencies; or (c) readmission for the management of osteomyelitis" [2]. This definition should be considered for future studies.

**Table 3. Pathogens identified in case series with minimum 10 presumed consecutive cases of pediatric chronic osteomyelitis including all bones.**

**Middle-high- and high-income countries**

| Author | Country | N | MSSA | MRSA | SA | *Pseudomonas* | *E. coli* | GAS | Other streptococci | Other pathogen [1] | Polymicrobial | No growth |
|---|---|---|---|---|---|---|---|---|---|---|---|---|
| McNeil [2] | US | 126[2] | 27 | 18 | | 9 | | | | 19 | 24 | 29 |
| Ulug [33] | Turkey | 21 | 9 | 3 | | 2 | 1 | | | 3 | 1 | 1 |
| Dieckmann [37] | Germany | 40 | | | 6 | | | | | 1 | | 33 |
| Unal [38] | Turkey | 22 | 6 | 4 | | | | | | 0 | 1 | 11 |
| Vogely [44] | Netherlands | 16 | | | 4 | | | | | 3 | | 9 |
| **Total** | | 225 | 42 (19%) | 25 (11%) | 10 (4%) | 11 (5%) | 1 (0.4%) | 0 | 0 | 26 (12%) | 26 (12%) | 83 (37%) |

**Middle-low- and low-income countries**

| Author | Country | | MSSA | MRSA | SA | *Pseudomonas* | *E. coli* | GAS | Other streptococci | Other pathogen | Polymicrobial | No growth |
|---|---|---|---|---|---|---|---|---|---|---|---|---|
| Bhattacharyya [15]] | India | 10 | 4 | 3 | | | | | | 1 | | 2 |
| Ellur [22] | India | 31 | 8 | 12 | | | | 2 | | 1 | | 8 |
| Mantero [32] | Kenya | 90 | 24 | 22 | | | 1 | 1 | | 1 | | 41[3] |
| Matzkin [11] | Pacific Islands[4] | 55 | 20 | 15 | | | | | | 0 | | 20[3] |
| Bahebeck [41] | Cameroon | 77 | | | 27 | 6 | 10 | | 4 | 9 | 14[3] | 7 |
| **Total** | | 263 | 56 (21%) | 52 (20%) | 27 (10%) | 6 (2%) | 11 (4%) | 3 (1%) | 4 (2%) | 12 (5%) | 14 (5%) | 78 (30%) |

Legend: *E. coli* – *Escherichia coli*; GAS – group A streptococcus; MRSA – methicillin resistant *Staphylococcus aureus;* MSSA – methicillin susceptible *Staphylococcus aureus;* SA - *Staphylococcus aureus* (susceptibilities not reported); US – United States.

[1]Anaerobes (N = 14); *Proteus* (N = 5); *Enterobacter* (N = 6); *Salmonella* (N = 5); coagulase negative staphylococci (N = 4); *Cutibacterium* (N = 1); *Klebsiella pneumoniae* (N = 1); *Brucella* (N = 1 – positive serology only); *Candida* (N = 1).

[2]total is higher than number of patients (N = 114) as not always clear which results were polymicrobial.

[3]numbers derived from the total number but not clearly stated in the manuscript.

[4]surgeries performed in United States.

It seems likely that the prognosis and optimal treatment vary by type of CO and by the volume of necrotic bone at presentation. Risk factors for post-AHO CO are not clear, but one study reported a higher risk if early bone ischemia was reported on MRI performed for AHO [14]. In the presence of orthopedic hardware, it is not clear how to differentiate acute from chronic osteomyelitis.

Optimal surgical management of CO is not clear from this review. The majority of cases had debridement, but some were cured without surgery. It seems likely that the need for debridement depends upon the volume of necrotic bone; recurrence may correlate with the volume of residual necrotic bone following surgery. For the US study where only 18% of cases had surgery [18], the diagnosis of CO was based on discharge diagnostic codes alone; it seems likely that some would not be classified as CO in other studies. Given the paucity of high-quality evidence that CO can be cured with antibiotics alone, debridement would seem to be indicated in most cases to remove necrotic bone and collect cultures.

In terms of antimicrobials, it is not clear from this review whether all patients require systemic antibiotics if adequate debridement is achieved. However, only 3 CO cases were managed without antibiotics [10] so clearly most clinicians consider them mandatory. The role of PO versus IV antibiotics is not clear. Very few patients received only PO antibiotics. However, a recent study reported that AHO can usually be managed with PO antibiotics alone [50], so perhaps IV antibiotics are only required if the patient is septic (which is rare with CO) or if absorption of or compliance with PO antibiotics is doubtful.

**Table 4. Treatment and outcomes of pediatric chronic osteomyelitis arranged by year and country economy.**

Upper-middle- and high-income economies

| Year | Author | Country | N | Managed without surgery (N) | Number of surgeries for management of CO | Duration of antibiotics following initial surgery | Antibiotics/ bioactive glass implanted at surgery | Recurrence [1] | Orthopedic sequelae [2] |
|---|---|---|---|---|---|---|---|---|---|
| 2024 | Al-alawi [14] | Oman | 5 | NR | NR | NR | NR | NR | NR |
| 2023 | Disch [18] | US | 343 | 280 (82%) | 1 (N=34); ≥2 (N=29) | NR | no | NR | NR |
| 2023 | Shi [19] 3 | China | 21 | 0 | 2 each | 2 weeks IV and 4 weeks PO | vancomycin in polymethyl methacrylate (N=21) | none at 21–61 months (mean 32) | 2 (10%) bone resorption; 1 (10%) refracture; 1 (10%) broken plate; 1 (10%) varus ankle |
| 2022 | Lazzeri [20] | Italy | 4 | 0 | NR | NR | bioactive glass | none at 3–5 years | NR |
| 2021 | Kojima [21] | Brazil | 5 | 0 | 1 (N=2); 2 (N=3) | NR | vancomycin in polymethyl methacrylate and bioactive glass (N=5) | none at 2.5 years | NR |
| 2021 | McNeil [2] | US | 114 | 34 (30%) | 1 (N=46); ≥2 (N=34) | mean IV 12 days (IQR 4–42); mean total 210 days (IQR 130–367; 8 received only IV, 65 received IV for < 14 days and 41 for 14+ days | no | 30 (26%) had treatment failure defined as signs/ symptoms of CO at last follow-up – risk increased if neurologic comorbidities or the presence of decubiti 4 | NR |
| 2019 | Andreacchio [24] | Italy | 12 | 0 | one each | mean 32 days IV (range: 14–90 days); mean 37 days PO (range: 14–60 days). | tobramycin in calcium sulphate (N=12) | none at 3–6 years | NR |
| 2018 | Munshi [25] | Fiji | 118 | 18 (15%) | NR | NR | no | ongoing illness or recurrent infection (N=31; 26%), complete resolution (N=72;61%); LTFU (N=15; 13%) | NR |
| 2018 | Rousset [9] | France | 8 | NR | NR | mean IV 26 days (range 5–90) and mean PO 22 days (range 10–42) | gentamicin in cement (N=8) | none at 0.5 to 5 years | 1 (12%) decrease range of motion; 1 (12%) 10 cm leg discrepancy |
| 2018 | Akyuz [27] 5 | Turkey | 3 | 0 | NR | all got TMP/SMX for over 2 years and were cured | no | none at 38–47 months | NR |
| 2015 | Costa [28] 6 | Portugal | 2 | 2 (100%) | 0 | 18 months PO | no | none | NR |

*(Continued)*

**Upper-middle- and high-income economies**

| Year | Author | Country | N | Managed without surgery (N) | Number of surgeries for management of CO | Duration of antibiotics following initial surgery | Antibiotics/ bioactive glass implanted at surgery | Recurrence[1] | Orthopedic sequelae[2] |
|---|---|---|---|---|---|---|---|---|---|
| 2015 | Shukrimi [29] | Malaysia | 3 | 2 of 3 (67%) | NR | NR | no | NR | 1 (33%) limb length discrepancy |
| 2011 | Ulug [33][7] | Turkey | 21 | 0 | NR | NR | no | NR | NR |
| 2010 | Bar-On [34] | Israel | 4 | 0 | 1 (N=2); 2 (N=2) | mean 6 weeks IV (range 3–13) and total of 16 weeks (range 10–37) | gentamicin in polymethyl methacrylate rods (N=4) | 1 (25%) required repeat surgery at 3 months but all well at 41 months | 1 (25%) pathological fracture |
| 2010 | Zeng [35] | China | 2 | 0 | 1 | Mean 16,5 days IV (range 14–19) and PO mean 22 days (range 14–30) | no | none at 6 and 11 months | NR |
| 2008 | Dieckmann [37] | Germany | 40 | 4 (10%) | NR | mean 11.1 days IV (range 3–27) and PO for mean 49 days (range 6–130) | a genta fleece (sulmycin implant® Innocoll) or a gentami chain (Septopal chains® Biomet) (N=37) – removed at 19–48 days | 6/40 (15%) recurred at 23 days to 3.5 years – one had a second recurrence 9 months after the first one | 5 12% had pain or reduced range of motion or minor deformities |
| 2006 | Unal [38] | Turkey | 22 | 0 | 1 (N=10); ≥2 (N=12) | minimum 6 weeks | no | none at mean 54 months | 4 (18%) diaphyseal curvature greater than 10 degrees; 1 (5%) non-union |
| 2005 | Beslikas [39][8] | Greece | 5 | 0 | 1 | 3 weeks IV and 2–3 months PO | no | 1 (20%) required repeat surgery at 3 months but all well at 3–10 years | none |
| 2004 | Yeargan [10][9] | US | 30 | 1 (3%) | 29 had 97 total | 0 to 20 weeks IV (mean 5.7 weeks); 4–24 weeks PO (mean 6.9); total duration 15.4 weeks in 1980s and 9.5 weeks in 1990s | no | none at mean 2.5 years (range 0–9 years) | 9 (13%) had leg length discrepancy of 2.5–3.5 cm |
| 2002 | Paley [40][10] | US | 4 | 0 | NR | NR | tobramycin and vancomycin in polymethyl methacrylate impregnated cement rod removed after 59,79, 94 or 212 days and replaced with regular rod in 3 of 4 cases | none at 38–47 months | NR |

(Continued)

Table 4. (Continued)

## Upper-middle- and high-income economies

| Year | Author | Country | N | Managed without surgery (N) | Number of surgeries for management of CO | Duration of antibiotics following initial surgery | Antibiotics/ bioactive glass implanted at surgery | Recurrence[1] | Orthopedic sequelae[2] |
|---|---|---|---|---|---|---|---|---|---|
| 2001 | Rasool [42] [11] | South Africa | 10 | 0 | 2 to 5 each | NR | no | none at 3 months to 6 yr | All 10 had sequalae as required joint fusions+/- or removal of all or part of calcaneus; 6 (60%) required modified shoes. |
| 2000 | Reinehr [43] | Germany | 10 | 5 (50%) | NR | 16 to 29 days (mean 21 days) IV- and 3-months PO | no | 1 (10%) recurred despite surgery initially | none |
| 1997 | Vogely [44] | Netherlands | 16 | 0 | one each | mean 20 days IV (range 8–32) and PO mean 25 days (range 21–45) | gentamicin beads (N=9) | 1 (6%) at mean 2.7 years (range 0.4 to 7.6) | 1 (6%) subtalar ankylosis in child with CO of calcaneus |
| 1994 | Lauschke [46] [12] | Namibia | 30 | 0 | 1 (N=27: 2 (N=2); 3 (N=1) | IV 3–4 weeks – do not mention PO | no | 4 (13%) at 24 months | 2 (7%) leg length discrepancies with decreased range of motion of hips |
| 1991 | Tudisco [48] | Italy | 26 | 10 (38%) | 1 (N=8); 2 (N=6); 3 (N=2) | 6 to 12 months (mean 8 months) | no | none at mean 23 years (range 11–41) but 24 (46%) LTFU | 4 (15%) limb length discrepancies |
| 1989 | Saighi Bouaouina [49] | Algeria | 46 | 0 | NR | 10 to 60 days | no | 7 (15%) relapsed within months of which 6 did not recur after a second surgery; followed 3 months to 20 years; 3 LTFU | 1 (2%) pathological fracture |

## Low and Lower-middle Income Economies

| Year | Author | Country | N | Managed without surgery (N) | Number of surgeries for management of CO | Duration of antibiotics following initial surgery | Antibiotics implanted at surgery | Recurrences | Orthopedic sequelae |
|---|---|---|---|---|---|---|---|---|---|
| 2024 | Bhattacharyya [15] | India | 10 | 0 | NR | IV mean 7 days and PO for about 14 days | gentamicin and vancomycin in calcium sulphate [N=10] | none | 1 (10%) non-union with limb shortening; 1 (10%) draining incision |
| 2024 | Peshin [16] | India | 100 | 0 | 1 (N=34); 2 (N=42); 3 (N=12); ≥4 (N=12) | 6 weeks (IV initially) | antibiotic-bone cement (N=18) – choice of antibiotic NR | NR | 16 (16%) did not show improvement of which 7(7%) required amputation |

*(Continued)*

**Upper-middle- and high-income economies**

| Year | Author | Country | N | Managed without surgery (N) | Number of surgeries for management of CO | Duration of antibiotics following initial surgery | Antibiotics/ bioactive glass implanted at surgery | Recurrence [1] | Orthopedic sequelae [2] |
|---|---|---|---|---|---|---|---|---|---|
| 2023 | Mulualem [17] | Ethiopia | 151 | NR | NR | NR | no | NR | 18 (12%) pathologic fracture; 5 (3%) angular deformity; 2 (1%) joint space narrowing; 2 (1%) ankylosis and effusion |
| 2021 | Ellur [22] [13] | India | 31 | 0 | NR | 4 to 7 days IV and 4 weeks PO | gentamicin and vancomycin in calcium sulphate (N = 34) | none at mean 42 months (range 28–70); 3 LTFU | NR |
| 2019 | Edson [23] | Uganda | 75 | NR | NR | NR | no | NR | 61 (82%) decreased range of motion |
| 2018 | Omoke [26] | Nigeria | 50 | NR | NR | NR | no | NR | NR |
| 2015 | Stevenson [12] – Beckles [13] [13] | Malawi | 167 | 0 | 1 (N = 110); 57 had 183 additional ones | usually 6 weeks PO but some also got IV | antibiotic spacers (N = 8) – details NR | none at minimum 12 months | 2 (1%) amputations for CO of calcaneus |
| 2014 | Wirbel [30] | Afghanistan/ Angola (surgery in Germany) | 27 | 0 | 2 to 8 each | 3-12 days IV and 6 weeks PO | no | recurred in first 6 months (N = 3; 11%); no recurrence (N = 15; 56%); LTFU (N = 9; 33%) | NR |
| 2013 | Ponio [31] | Philippines | 80 | 6 (8%) | NR | often 2–3 weeks IV and total of 4–6 weeks | no | none but no follow-up documented | NR |
| 2011 | Mantero [32] | Kenya | 96 | 0% | one each [14] | usually 6 weeks total with switch to PO when biomarkers normalizing | no | 11 (15%) by 12 months | none |
| 2009 | Akakpo-Numado [36] | Togo | 23 | NR | NR | NR | NR | NR | 2 (9%) pathologic fractures |
| 2005 | Matzkin [11] | Pacific Islands (surgery in US) | 55 | 7 (13%) | mean of 1.3 each (range 0–6) | mean 28 days IV and 107 days PO and 135 days total | no | NR | NR |

(Continued)

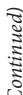

**Table 4.** (Continued)

**Upper-middle- and high-income economies**

| Year | Author | Country | N | Managed without surgery (N) | Number of surgeries for management of CO | Duration of antibiotics following initial surgery | Antibiotics/ bioactive glass implanted at surgery | Recurrence[1] | Orthopedic sequelae[2] |
|------|--------|---------|---|------|------|------|------|------|------|
| 2002 | Bahebeck [41][15] | Cameroon | 49 | 0 | 1 (N = 16); 2 (N = 21); 3 (N = 12) | 8+weeks | no | 2 (4%) required repeat surgery in first 6 weeks; 45 (90%) well at mean 14 months but one had persistent drainage the cleared; 4 (8%) LTFU | NR |
| 1995 | Bassey [45] | Nigeria | 41 | 0 | 1 (N = 33); ≥2 (N = 8) | NR | no | none at 3 years | 3 (7%) pathologic fractures; 6 (25%) stiff joints; 2 (5%) limb length discrepancy |
| 1991 | Onuba [47] | Zimbabwe | 20 | 0 | NR | 2 days IV and 6 weeks PO | no | NR | NR |

Legend– CO – chronic osteomyelitis; LTFU – lost to follow-up; NR – not reported; PO – per os.

[1]Some studies report only late recurrences while others report recurrences before initial therapy completed.

[2]Structural or functional.

[3]Only long bone cases included.

[4]Mean duration antibiotics longer in those who failed treatment than in those who did not (295 days [IQR: 180–394] vs. 180 days [IQR: 97–356], P=0.03.

[5]Only sternal cases included.

[6]Only Q fever cases included.

[7]Only cases with sinus tracts were included as the primary outcome was comparison of sinus and bone cultures.

[8]Only pelvic cases included.

[9]Only tibial cases included.

[10]Only included cases with infected intramedullary nail.

[11]Only calcaneal cases included.

[12]Only included hematogenous cases that required surgery.

[13]hematogenous cases only.

[14]irrigation of the medullary canal performed by in-out system for 7 days post-operatively.

[15]Only included cases that required surgery.

It was not possible to analyze the efficacy of specific antibiotics for CO as the choice of antibiotics was often not reported in detail, including in cases with recurrences. Whenever practical, cultures from bone should be obtained prior to administration to guide antibiotic choice. Assuming another pathogen was not previously detected from an operative specimen, empiric antibiotics should target *S. aureus* (with methicillin resistant *S. aureus* coverage if the local incidence is high) as this was the pathogen in about three-quarters of cases, recognizing that other pathogens may play an important role in types of CO other than post-AHO CO [2]. When cultures are negative, molecular detection methods should be considered [51]. The role of combination antibiotics is not clear. Rifampin has excellent bone penetration so is sometimes added to other antibiotics to treat *S. aureus* [52]. A randomized controlled trial reporting a trend towards improved outcomes with the addition of rifampin to 42 days IV nafcillin in 18 adults with CO without orthopedic hardware [53]. There are discordant results regarding the efficacy of rifampin for other device-related infections in adults [54]. Rifampin is yet to be studied in pediatric CO.

Local delivery of antibiotics was reported in 12 studies, through antibiotic loaded PMMA cement beads or spacers that eventually need to be removed or through antibiotic loaded calcium sulphate which is biodegradable. It is not clear whether PMMA or calcium sulfate interferes with healing or whether nephrotoxicity ever occurs. Efficacy is impossible to establish from the 12 studies as there was never a control group. An observational study that was excluded from the current review as 29% of cases were adults reported improved outcomes with implanted gentamicin beads in tibial CO than in unmatched controls with gentamicin rinses delivered via closed lavage [55]. Regarding the choice of antibiotics for local delivery, vancomycin is likely to cover *S. aureus*. Although gentamicin and tobramycin are synergistic with beta lactams for treatment of methicillin-susceptible *S. aureus* (MSSA), one would never use them as monotherapy for MSSA. A Nigerian study reported use of non-commercial ceftriaxone beads (which would cover MSSA) in adults and children [56].

The total duration of antibiotics varied markedly in this review. A 2010 systematic review that differed from the current review in that they included sub-acute osteomyelitis reported markedly varying durations of IV and PO antibiotics with no relationship between duration and treatment failure [57]. This fits with the results of the current study.

A potential new intervention is an injectable in situ-forming depot antibiotics delivery system which appears hopeful in animal models [58]. Success with hyperbaric oxygen has been reported in adults [59] and in one child [60].

A key limitation of this review is the inconsistent definitions applied for CO which made it impossible to combine study results. One study required only 7 days of symptoms [46] which most experts would not consider to be CO. However, our protocol specified inclusion of studies that the authors considered to be CO; it would introduce bias to arbitrarily exclude studies. It is possible that some included patients had AHO or CNO. Organism reported to be pathogens could have been contaminants. Incomplete reporting of types of CO and antibiotic regimens limited our ability to correlate outcomes with management decisions. Only published studies were screened.

## Conclusion

The results of this review should be applied to guide further study of CO. The first step is to settle on a definition. As previously stated, we favor the McNeil definition [2]. Debridement would seem to be indicated unless i) CO involves a small bone, ii) there is concern that debridement will contribute to bony instability, or iii) the lesion is too small to readily find. There is a need to compare outcomes with and without initial IV versus PO antibiotics. Beta lactams are typically used for AHO, but clindamycin, ciprofloxacin and trimethoprim-sulfamethoxazole have better bone penetration so should be compared to beta lactams for CO [52], with or without the addition of rifampin. The optimal duration of antibiotics probably depends upon the volume of residual necrotic bone post-debridement. Spellberg advocates a maximum 6-week course in adults [52], but a longer course can perhaps be justified if adequate debridement was not achieved. Given the rarity and heterogeneity of CO, multicenter randomized controlled trials may not be practical, so the next step could involve applying and studying a protocol in multiple centers. Hopefully advances in the next decade will improve the prognosis of pediatric CO worldwide.

## Supporting information

**S1 File. Search strategy.** This is the strategy for searching the literature.
(PDF)

**S2 File. PRISMA checklist.** This is the completed PRISMA checklist.
(DOCX)

**S3 File. Raw data.** This is the data as it was entered into REDCap.
(DOCX)

## Author contributions

**Conceptualization:** Joan L. Robinson, Liz Dennett.

**Data curation:** Joan L. Robinson, Deema Gashgarey, Nourah Alruqaie, Liz Dennett.

**Formal analysis:** Joan L. Robinson.

**Methodology:** Joan L. Robinson.

**Writing – original draft:** Joan L. Robinson.

**Writing – review & editing:** Joan L. Robinson, Deema Gashgarey, Nourah Alruqaie, Liz Dennett, M. Elizabeth Pedersen.

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
