## [Decision Letter · Decision Letter 0]

16 Oct 2025

Dear Dr. Robinson,

Thank you for submitting your manuscript to PLOS ONE. After careful consideration, we feel that it has merit but does not fully meet PLOS ONE’s publication criteria as it currently stands. Therefore, we invite you to submit a revised version of the manuscript that addresses the points raised during the review process.

We look forward to receiving your revised manuscript.

Kind regards,

Nan Jiang

Academic Editor

PLOS ONE

Journal Requirements:

2. Please include captions for your Supporting Information files at the end of your manuscript, and update any in-text citations to match accordingly. Please see our Supporting Information guidelines for more information: http://journals.plos.org/plosone/s/supporting-information .

Reviewers' comments:

Reviewer's Responses to Questions

**Comments to the Author**

1. Is the manuscript technically sound, and do the data support the conclusions?

Reviewer #1: Yes

Reviewer #2: Partly

Reviewer #3: Partly

2. Has the statistical analysis been performed appropriately and rigorously?

Reviewer #1: I Don't Know

Reviewer #2: Yes

Reviewer #3: N/A

3. Have the authors made all data underlying the findings in their manuscript fully available?

Reviewer #1: Yes

Reviewer #2: Yes

Reviewer #3: Yes

4. Is the manuscript presented in an intelligible fashion and written in standard English?

Reviewer #1: Yes

Reviewer #2: Yes

Reviewer #3: Yes

Reviewer #1: The topic is both clinically important and timely, considering the ongoing challenges in diagnosing and managing chronic bone infections in pediatric patients. The authors have made a commendable effort to summarize existing research and shed light on current practices and gaps in care.

*Strengths:*

-The manuscript is well-organized and adheres to PRISMA guidelines.

-Inclusion criteria are clearly stated, and the literature search appears thorough.

-The discussion effectively emphasizes the variation in treatment approaches and the urgent need for standardized protocols.

*Areas for Improvement:*

-The paper lacks formal statistical comparisons between treatment methods, outcomes, or regional differences.

-The conclusion is brief and not presented as a separate section. Creating a dedicated conclusion section would enhance the structure.

-The overall organization could be improved for better readability.

*Minor Suggestions*

-There are minor grammatical issues throughout the manuscript. A round of professional copyediting is recommended to improve clarity and flow.

-Table style should be consistent across the manuscript.

-Numbering styles (Roman numerals, alphabetical, numeric) vary—please choose one format and apply it consistently.

Reviewer #2: 1.The topic is clinically meaningful, and the review collects a good amount of pediatric chronic osteomyelitis data. However, the article would benefit from a clearer presentation of the review process. I suggest describing the search strategy, inclusion/exclusion criteria, and quality assessment more transparently, ideally following PRISMA 2020 standards.

2.The definitions of chronic osteomyelitis used across studies are highly variable. It would strengthen the paper to discuss how this heterogeneity might affect data interpretation and to recommend a unified diagnostic standard for future studies.

3.The results section is mostly descriptive. Some quantitative summaries (e.g., percentages of male patients, main infected bones, pathogen distribution, recurrence rates) or even simple comparisons between high- and low-income countries would make the findings more informative.

4.The discussion on antibiotic therapy could be expanded. Please summarize the main types, routes, and durations of antibiotics used, and comment on how these relate (or fail to relate) to recurrence or outcomes.

5.The section on local antibiotic delivery is interesting but brief. Consider summarizing which materials (PMMA, calcium sulfate, bioactive glass) were used, their reported success, and possible risks or limitations.

6.The conclusion could be a bit more specific, highlighting a few concrete takeaways — for example, the need for standardized definitions, multicenter studies, and evaluation of antibiotic duration and local delivery methods.

7.Minor points: language is generally clear but can be slightly shortened in places; ensure all tables and figures include clear titles, sample sizes, and consistent units.

Reviewer #3: The epidemiology and management of chronic OM in pediatrics:

Conducting a systematic review that consolidates multiple acute conditions progressing to chronic osteomyelitis and its sequelae into a unified treatment consensus is a significant challenge.

The authors have not formulated a precise structured question to be answered by the studies. A structured question for a systematic review should typically follow a recognized framework, most commonly PICO.

The objective of the study, presented by the authors, was to review the literature on managements and outcomes of pediatric CO and summarize the demographics, pathogens, treatments offered and outcomes.

There are some large limitations:

Intervention/Management is vague – not clearly presented in the question/objective and the outcome is not specified.

The classification of low/lower-middle income countries and upper-middle- or high-income countries is not presented as an objective of study.

The authors introduce the present study by referencing a 23-year-old publication by Ramos (2002), which uses a non-standard medical classification for osteomyelitis and the publication is difficult to get a hold of. The definition used in the introduction post-acute hematogenous osteomyelitis is not a standard medical classification on its own.

The authors do not clearly classify osteomyelitis, nor do they assist the reader in distinguishing between chronic infectious osteomyelitis and autoinflammatory bone diseases such as CRMO/CNO in the pediatric population. This distinction should be clarified. Additionally, the most common causes and background of chronic infectious osteomyelitis could be more thoroughly presented.

A recently published book, Pediatric Musculoskeletal infections edited by Belthur et al (2022) (Springer Cham), covers epidemiology and current concepts of pediatric musculoskeletal infections. This book is not mentioned in the background.

Figure 1:

The total sum of the studies included in the review is according to Figure 1, is 41. When subtracting removed excluded, and not retrieved studies, the sum is 42.

Supplement – search methodology

The number of publications is consistent across MEDLINE, Scopus, and Embase, but differs for CINAHL, n=144 in Figure 1 vs. n=137 in the Supplement.

**Do you want your identity to be public for this peer review?** For information about this choice, including consent withdrawal, please see our Privacy Policy

Reviewer #1: **Yes: ** Shibarjun Mandal

Reviewer #2: No

Reviewer #3: No

---

## [Author Response · Author response to Decision Letter 1]

4 Nov 2025

Reviewer #1: The topic is both clinically important and timely, considering the ongoing challenges in diagnosing and managing chronic bone infections in pediatric patients. The authors have made a commendable effort to summarize existing research and shed light on current practices and gaps in care.

*Strengths:*

-The manuscript is well-organized and adheres to PRISMA guidelines.

-Inclusion criteria are clearly stated, and the literature search appears thorough.

-The discussion effectively emphasizes the variation in treatment approaches and the urgent need for standardized protocols.

*Areas for Improvement:*

-The paper lacks formal statistical comparisons between treatment methods, outcomes, or regional differences.

Response: Thanks for pointing out this important omission. We added the following explanation to the methods section: “The initial plan was to perform a meta-analysis of outcomes but this was not conducted due to i) the heterogeneity of CO definitions ii) the fact that often minimal data were provided on the initial management of cases that recurred and iii) the markedly variable durations and completeness of follow-up.”

-The conclusion is brief and not presented as a separate section. Creating a dedicated conclusion section would enhance the structure.

Response: We added the following as a conclusion:

“The results of this review should be applied to guide further study of CO. The first step is to settle on a definition. As previously stated, we favor the McNeil definition (2). Debridement would seem to be indicated unless i) CO involves a small bone, ii) there is concern that debridement will contribute to bony instability, or iii) the lesion is too small to readily find. There is a need to compare outcomes with and without initial IV antibiotics. Beta lactams are typically used for AHO but clindamycin, ciprofloxacin and trimethoprim-sulfamethoxazole have better bone penetration so should be compared to beta lactams for CO (52), with or without the addition of rifampin. The optimal duration of antibiotics probably depends upon the volume of residual necrotic bone post-debridement. Spellberg advocates a maximum 6-week course in adults (52) , but a longer course can perhaps be justified if adequate debridement was not achieved. Given the rarity and heterogeneity of CO, multicenter randomized controlled trials may not be practical so the next step could involve applying and studying a protocol in multiple centers. Hopefully advances in the next decade will improve the prognosis of pediatric CO worldwide.”

-The overall organization could be improved for better readability.

Response: We tried to make the entire manuscript clearer.

*Minor Suggestions*

-There are minor grammatical issues throughout the manuscript. A round of professional copyediting is recommended to improve clarity and flow.

-Table style should be consistent across the manuscript.

-Numbering styles (Roman numerals, alphabetical, numeric) vary—please choose one format and apply it consistently.

Response: We addressed these concerns and in particular dealt with empty cells in the tables.

Reviewer #2: 1.The topic is clinically meaningful, and the review collects a good amount of pediatric chronic osteomyelitis data. However, the article would benefit from a clearer presentation of the review process. I suggest describing the search strategy, inclusion/exclusion criteria, and quality assessment more transparently, ideally following PRISMA 2020 standards.

Response: We tried to more closely follow the PRISMA 2020 standards for both the abstract and the body of the manuscript.

2.The definitions of chronic osteomyelitis used across studies are highly variable. It would strengthen the paper to discuss how this heterogeneity might affect data interpretation and to recommend a unified diagnostic standard for future studies.

Response: We now state: “A key limitation of this review is the inconsistent definitions applied for CO which made it impossible to combine study results.” We now specify which CO definition we think that future studies should consider using.

3.The results section is mostly descriptive. Some quantitative summaries (e.g., percentages of male patients, main infected bones, pathogen distribution, recurrence rates) or even simple comparisons between high- and low-income countries would make the findings more informative.

Response: We added more actual data to the results section in both the abstract and the body of the manuscript.

4.The discussion on antibiotic therapy could be expanded. Please summarize the main types, routes, and durations of antibiotics used, and comment on how these relate (or fail to relate) to recurrence or outcomes.

Response: The duration of antibiotics was too variable to summarize in the text, but where provided is now added to Table 4. Unfortunately, there was rarely sufficient data for individual patients with recurrences to know what antibiotics they received. We therefore state: “A statistical comparison of outcomes was not conducted given the heterogeneity of definitions and incomplete descriptions of management, but there is no obvious link between the duration or route of delivery of antibiotics and outcomes.”

5.The section on local antibiotic delivery is interesting but brief. Consider summarizing which materials (PMMA, calcium sulfate, bioactive glass) were used, their reported success, and possible risks or limitations.

Response: This data are now provided in Table 4. We summarize efficacy as follows: ” All report initial success, but one that used gentamicin reported that 6 of 40 had recurrences at 23 days to 3.5 years with one having a second recurrence 9 months later (37)”.

6.The conclusion could be a bit more specific, highlighting a few concrete takeaways — for example, the need for standardized definitions, multicenter studies, and evaluation of antibiotic duration and local delivery methods.

Response: Thanks for this suggestion. We added the following specific suggestions to the conclusion: “The results of this review should be applied to guide further study of CO. The first step is to settle on a definition. As previously stated, we favor the McNeil definition (2). Debridement would seem to be indicated unless i) CO involves a small bone, ii) there is concern that debridement will contribute to bony instability, or iii) the lesion is too small to readily find. There is a need to compare outcomes with and without initial IV antibiotics. Beta lactams are typically used for AHO but clindamycin, ciprofloxacin and trimethoprim-sulfamethoxazole have better bone penetration so should be compared to beta lactams for CO (52), with or without the addition of rifampin. The optimal duration of antibiotics probably depends upon the volume of residual necrotic bone post-debridement. Spellberg advocates a maximum 6-week course in adults (52) , but a longer course can perhaps be justified if adequate debridement was not achieved. Given the rarity and heterogeneity of CO, multicenter randomized controlled trials may not be practical so the next step could involve applying and studying a protocol in multiple centers. Hopefully advances in the next decade will improve the prognosis of pediatric CO worldwide.”

7.Minor points: language is generally clear but can be slightly shortened in places; ensure all tables and figures include clear titles, sample sizes, and consistent units.

Response: We tried to be more concise and to ensure that the tables and figures contained all relevant information.

Reviewer #3: The epidemiology and management of chronic OM in pediatrics:

Conducting a systematic review that consolidates multiple acute conditions progressing to chronic osteomyelitis and its sequelae into a unified treatment consensus is a significant challenge.

The authors have not formulated a precise structured question to be answered by the studies. A structured question for a systematic review should typically follow a recognized framework, most commonly PICO.

Response: Thanks for this suggestion. We did not study a specific intervention so it is difficult to turn our research question into a traditional PICO. However, we now start the methods section with: “This was a systematic review of the interventions and outcomes of patients up to 17 years of age with CO. The primary outcome was the recurrence rate. We analyzed cases separately in upper-middle- or high-income countries versus low or lower-middle income countries as the latter more commonly have a markedly delayed presentation.”

The objective of the study, presented by the authors, was to review the literature on managements and outcomes of pediatric CO and summarize the demographics, pathogens, treatments offered and outcomes.

There are some large limitations:

Intervention/Management is vague – not clearly presented in the question/objective and the outcome is not specified.

The classification of low/lower-middle income countries and upper-middle- or high-income countries is not presented as an objective of study.

Response: See above.

The authors introduce the present study by referencing a 23-year-old publication by Ramos (2002), which uses a non-standard medical classification for osteomyelitis and the publication is difficult to get a hold of. The definition used in the introduction post-acute hematogenous osteomyelitis is not a standard medical classification on its own.

Response: Thanks for pointing out these problems. We changed the introduction to use the definition of CO that was used in the 2021 Infectious Diseases Society of America guidelines.

The authors do not clearly classify osteomyelitis, nor do they assist the reader in distinguishing between chronic infectious osteomyelitis and autoinflammatory bone diseases such as CRMO/CNO in the pediatric population. This distinction should be clarified. Additionally, the most common causes and background of chronic infectious osteomyelitis could be more thoroughly presented.

Response: Thanks for this suggestion. Because this is a systematic review rather than a narrative review, we did not think that readers would go to this manuscript for background information on CO. Much of the background information comes out in our results and discussion. However, we did add more information on CNO. If the editor or reviewer disagrees with this decision, we could add more background information to the introduction.

A recently published book, Pediatric Musculoskeletal infections edited by Belthur et al (2022) (Springer Cham), covers epidemiology and current concepts of pediatric musculoskeletal infections. This book is not mentioned in the background.

Response: Thanks for this suggestion but we were not able to access this book without purchasing it.

Figure 1:

The total sum of the studies included in the review is according to Figure 1, is 41. When subtracting removed excluded, and not retrieved studies, the sum is 42.

Supplement – search methodology

The number of publications is consistent across MEDLINE, Scopus, and Embase, but differs for CINAHL, n=144 in Figure 1 vs. n=137 in the Supplement.

Response: Thanks for noticing these two errors which have been corrected.

---

## [Decision Letter · Decision Letter 1]

10 Nov 2025

The epidemiology and management of chronic osteomyelitis in pediatrics – A systematic review

PONE-D-25-46757R1

Dear Dr. Robinson,

We’re pleased to inform you that your manuscript has been judged scientifically suitable for publication and will be formally accepted for publication once it meets all outstanding technical requirements.

Kind regards,

Nan Jiang

Academic Editor

PLOS ONE

Reviewers' comments:

Reviewer's Responses to Questions

**Comments to the Author**

Reviewer #1: All comments have been addressed

Reviewer #2: All comments have been addressed

2. Is the manuscript technically sound, and do the data support the conclusions?

Reviewer #1: Yes

Reviewer #2: Yes

3. Has the statistical analysis been performed appropriately and rigorously?

Reviewer #1: N/A

Reviewer #2: Yes

4. Have the authors made all data underlying the findings in their manuscript fully available?

Reviewer #1: (No Response)

Reviewer #2: Yes

5. Is the manuscript presented in an intelligible fashion and written in standard English?

Reviewer #1: Yes

Reviewer #2: Yes

Reviewer #1: (No Response)

Reviewer #2: (No Response)

**Do you want your identity to be public for this peer review?** For information about this choice, including consent withdrawal, please see our Privacy Policy

Reviewer #1: **Yes: ** Shibarjun Mandal

Reviewer #2: No

---

## [Editor Report · Acceptance letter]

PONE-D-25-46757R1

PLOS ONE

Dear Dr. Robinson,

I'm pleased to inform you that your manuscript has been deemed suitable for publication in PLOS ONE. Congratulations! Your manuscript is now being handed over to our production team.

Kind regards,

on behalf of

Dr. Nan Jiang

Academic Editor

PLOS ONE